# Up in the Air: Evidence of Dehydration Risk and Long-Haul Flight on Athletic Performance

**DOI:** 10.3390/nu12092574

**Published:** 2020-08-25

**Authors:** Damir Zubac, Alex Buoite Stella, Shawnda A. Morrison

**Affiliations:** 1Faculty of Kinesiology, University of Split, Teslina 6, 21000 Split, Croatia; 2Science and Research Centre Koper, Institute for Kinesiology Research, 6000 Koper, Slovenia; 3Clinical Unit of Neurology, Department of Medicine, Surgery and Health Sciences, Cattinara University Hospital ASUGI, University of Trieste, Strada di Fiume, 447, 34149 Trieste, Italy; alex.buoitestella@gmail.com; 4Faculty of Sport, University of Ljubljana, Gortanova 22, 1000 Ljubljana, Slovenia; shawnda.morrison@fsp.uni-lj.si

**Keywords:** athletic performance, jet lag syndrome, fluid intake, hypohydration

## Abstract

The microclimate of an airline cabin consists of dry, recirculated, and cool air, which is maintained at lower pressure than that found at sea level. Being exposed to this distinctive, encapsulated environment for prolonged durations, together with the short-term chair-rest immobilization that occurs during long-haul flights, can trigger distinct and detrimental reactions to the human body. There is evidence that long-haul flights promote fluid shifts to the lower extremity and induce changes in blood viscosity which may accelerate dehydration, possibly compromising an athlete’s potential for success upon arrival at their destination. Surprisingly, and despite several recent systematic reviews investigating the effects of jet lag and transmeridian travel on human physiology, there has been no systematic effort to address to what extent hypohydration is a (health, performance) risk to travelers embarking on long journeys. This narrative review summarizes the rationale and evidence for why the combination of fluid balance and long-haul flight remains a critically overlooked issue for traveling persons, be it for health, leisure, business, or in a sporting context. Upon review, there are few studies which have been conducted on actual traveling athletes, and those that have provide no real evidence of how the incidence rate, magnitude, or duration of acute dehydration may affect the general health or performance of elite athletes.

## 1. Study Rationale

To compete internationally, elite athletes are frequently exposed to long-distance air travel, often across multiple time zones. This results in a cluster of acute, detrimental, health-related symptoms, commonly known as jet lag. Although there have been a number of major review articles on jet lag published over the past ~10 years which suggest avoiding dehydration or advocating “staying hydrated” whilst flying [1,2,3,4], none discuss the magnitude of expected levels of dehydration to be experienced over a given flight duration, or whether athletes could be at greater risk for becoming hypohydrated compared to other populations. Thus, there remains a need to determine the influence transmeridian air travel has on the physical performance outcomes of athletes.

## 2. PART ONE: Mechanisms of Hydration State Changes during Long-Haul Flights

### 2.1. The Airline Cabin Environment

When flying, cabin environmental characteristics are different from the normal conditions on land in that cabin air is characterized by reduced oxygen partial pressure and lower humidity [5]. Additional factors which may impact human physiology, and have yet to be fully investigated, include: the amount of air flow, uneven cabin air distribution, and the proportion of recycled air in the aircraft environment [6]. During flight, the cabin is a ventilated, enclosed environment that is controlled by the Environmental Control System (ECS) which provides the air to the passengers and crew. Pressurization of the cabin is necessary since at most flight altitudes (e.g., 11,000 m), atmospheric pressure is associated with an oxygen partial pressure nearly incompatible with human life. As such, the cabin pressure can range from a maximum of 101 kPa at sea level, to a minimum of 75 kPa in flight, corresponding to an altitude of ~2440 m [7,8]. Because of the density of the passengers and the limited volume inside the aircraft, cooling is usually necessary to prevent uncomfortable heating of the cabin environment. Circulated air is usually maintained to around 23 °C by supplying the cabin with air at minimum 10 °C [7]. Control of the relative humidity of the air inside the cabin is one of the main tasks of the ECS, since humidity may affect both passengers’ comfort and the structural integrity of the aircraft. Typically, at cruise altitudes, relative humidity is maintained at 10–20% by removing moisture from the cabin air; this is to avoid structural damages to the aircraft, [7].

Taken together, these factors (i.e., dry air, low O_2_ pressure) may result in an increased insensible water loss, including evaporation of water at the skin surface [4] and respiratory water loss as a product of the combination of increased ventilation (in particular, changes to tidal volume) [5] and increased need to humidify the air that enters the lungs [1,2,6]. Indeed, increasing water intake by 15–20 mL has been suggested for each hour of flight [3]; however, this recommendation is likely insufficient to prevent dehydration, since it is reported that resting ventilatory water losses can increase from 160 mL/hour to 360 mL/hour when relative humidity decreases from 60% to 12%, a drop consistent with airline cabin environments. These calculations assume an ambient temperature of ~21 °C and ventilation rate of 7 L/hour [4,9,10]. It remains that prescribing fluid intake of only 15–20 mL/hour may vastly underestimate the actual hydration needs of those flying for extended periods of time.

### 2.2. Fluid Intake during Long-Haul Flights

Due to the above-mentioned factors, it may be speculated that increased water losses are probable during flight, and optimal fluid intake should compensate these expected losses in order to avoid becoming hypohydrated [1,2]. However, only a handful of studies have investigated the fluid intake habits of people during long-haul flights.

For example, one study investigated 10 healthy volunteers during a 10 h simulated flight at 2800 m, and found that plasma volume decreased by 6% to 9%; the authors ascribed these differences to increased insensible water loss and decreased fluid intake that may have contributed to in-flight hypovolemia [11]. Total fluid intake consisted of ~960 mL of nonalcoholic fluid. Pure water was compared to a high-sodium–low-carbohydrate drink, or to a low-sodium–high-carbohydrate drink, with the findings that plasma volume was better preserved using the electrolyte–carbohydrate solutions, possibly due to the lower urine output [11] observed with that group. Despite the few studies assessing hydration status during long-haul flights, some findings are available for aviators’ fluid balance during military flights. Levkovsky et al. [12] investigated the hydration of 48 aviators who participated in a total of 104 training flights. The authors found that their mean fluid loss rate was ~465 mL/h, as evaluated by the difference in body weight [12]; nevertheless, this data should be carefully interpreted in the context of athletes traveling during a long-haul flight since the metabolic requirements and protective equipment of military aviators are far different from the resting conditions of commercial flight passengers.

In another study, Silva et al. [13] retrospectively queried the fluid intake habits of 94 male kite surfers by asking them to recall the longest recent flight the athlete took for a competition [13]. During their recalled flight, it was reported that 66% of the kite surfers self-reported drinking “some kind of fluid” during the flight, and the majority (51%) self-reported consuming less than 0.5 L of water. The authors did not report any association in fluid intake with flight duration (travel distance varied greatly between 14.489 ± 3629 km and 2264 ± 781 km for the two subgroups). Although the research emphasis of these works focuses on amount of fluid intake during flights, overhydration has also been mentioned to be avoided due to the possible disruptions frequently getting up to void the bladder will have on sleep during the flight [14].

### 2.3. Effect of Jet Lag and Long-Haul Flight on Gastrointestinal and Renal Function

The expression of key genes in peripheral organs throughout the day is orchestrated by a central pacemaker in the brain, namely the suprachiasmatic nucleus (SCN) of the anterior hypothalamus, that takes cues from cycles of light and dark, food, hormone levels, or from the metabolic status of the individual (e.g., exercise) [15]. Mechanistically, arginine vasopressin/V1 receptor signaling in the SCN plays a critical role in the resilience of the circadian clock to jet lag [16,17]; however, to the best of the authors’ knowledge, no study has investigated the associated between hydration status, vasopressin, and jet lag symptoms.

Researchers do know that circadian rhythms control several gastrointestinal functions, including gastric enzyme and fluid production, small intestine nutrient absorption, and gastric and gut motility. Under normal physiological conditions, the gastrointestinal system is quiescent during the night, rapidly increasing activity after awakening and throughout the day [18]. Amongst the primary associations between circadian rhythm and gastrointestinal function, motility in the different tracts of the gastrointestinal system is one of the key physiological functions that is altered due to circadian cycle disruption [19]. Melatonin, the neuroendocrine clock factor produced by the pineal gland, is highly present in the gastrointestinal tissue and therefore is implicated in digestive function [20]. Circadian disruption shifts the timing of eating and normal gastrointestinal functions, such as secretions, enzyme activity, intestinal motility, and the rate of nutrient absorption, often resulting in pathological conditions such as abdominal pain, constipation, and diarrhea [19]. Suffering from these side effects would have a direct influence on fluid intake behavior and overall fluid balance of any traveler. If nutrient absorption and gastric and gut motility are affected by circadian disruption, then it may be that the athlete cannot absorb as much water as is needed to remain well hydrated, or they may simply lack the appetite to consume additional liquids due to actual gut discomfort.

Appetite loss and gastrointestinal distress are often reported because of jet lag, and dietary interventions may help modulate these symptoms, although further research is needed to definitively establish the role of diet on jet lag adaptation [21]. If appetite loss is a direct consequence of circadian rhythm disruption, then gastrointestinal distress may also be a consequence of poor hygiene practices or food/water safety in the foreign location. Traveler’s diarrhea, especially when traveling to another country where food contamination may be more likely to occur, is a common feature in as many as 60% of athletes who travel internationally [22]. This kind of disruption can obviously significantly impair performance. On the other hand, reducing fluid intake during the flight to avoid frequent trips to the bathroom, or to reduce the risk of consuming fluids that may be contaminated, may result in constipation. In either case, it is important for the athlete not to overcorrect their fluid balance situation, so to speak. To the best of the authors’ knowledge, there is a scarcity of studies investigating renal function during long-haul flights or circadian rhythm disruption, with some results present for chronic desynchronizations. In hamsters, for example, prolonged circadian disorganization may be associated with proteinuria, tubular dilation, and cellular apoptosis [23]. More recently, in humans, daytime napping has been suggested as being associated with negative effects on renal function, namely hyperfiltration and microalbuminuria [24,25]. How these issues directly link to the traveling athlete remains to be determined. Certainly, being aware that it is important to continue to consume fluids regularly during long-haul flights is important to convey to traveling athletes.

### 2.4. Other Medical Issues of Long-Haul Flights: Venous Thromboembolism

Among the medical issues that may be present during long-haul flights, and which may undoubtedly benefit from a proper hydration protocol, is venous thromboembolism (VTE), a condition which shows an increased risk on flights greater than >8 h duration. Indeed, there is some risk present already after 4 h [8]. A population-based study including 9000 business travelers showed an absolute risk for venous thromboembolism for one in every 4656 flights [26]. This increased risk is linked to the reduced mobility during the flight [27]. Dehydration has been listed as one of the factors that may predispose oneself to a higher risk of VTE [9,27]. Certainly, it has been established that as a consequence of dehydration, hemoconcentration and increased blood viscosity may lead to hypercoagulability which may be present also in athletes [28,29], although conflicting results are reported [30]. A further risk factor to VTE may be represented by the fluid shifts that occur during prolonged chair rest mobility, especially since dehydration was found to increase lower limb edema in otherwise healthy people during long simulated flights [31,32].

In a particularly elegant study, one investigation looked at whether fluid loss occurred more in individuals with coagulation activation after air travel and compared the responses to participants without coagulation activation [30]. The secondary aim of the study was to examine fluid losses that occurred during actual air travel. In their crossover study, 71 healthy volunteers were exposed to eight hours of air travel, eight hours of immobilization in a cinema, and a daily-life control situation. Markers of fluid loss (hematocrit, serum osmolality, and albumin) and of coagulation activation were assessed before and after each exposure. There were 11 volunteers with, and 55 volunteers without, coagulation activation during the flight. The authors found that fluid loss was not different in volunteers with an activated clotting system from those without (difference between groups in hematocrit: −0.6%, 95% confidence interval [CI]: −1.9 to 0.6). On a group level, mean hematocrit values decreased during all three exposures, however, in some individuals, it increased; this occurred in more participants during the flight scenario (34%; 95% CI 22 to 46) than during the daily-life situation (19%; 95% CI 10 to 28). Ultimately, these findings do not support the hypothesis that fluid loss itself contributes to thrombus formation during air travel. However, one limitation to this study was that urinary hydration markers were not assessed so a complete picture of hydration status was unavailable.

Based on all the mechanisms described above, increased fluid intake is often recommended as a valid prophylaxis for VTE; however, these suggestions seem to be based more on common sense than on actual evidence [8,33]. On a final note, the type of fluid ingested may also play a role in VTE prevention. A research letter described their study which investigated the effects of an electrolyte and carbohydrate beverage (ECB) compared to water on 40 healthy men during a 9 h flight. The authors found that the ECB increased plasma volume compared to water, which was also associated with reduced urine output and positive fluid balance. Interestingly, elevated foot blood viscosity was lower in the ECB group, an encouraging finding that using such fluids during long-haul flights do contribute to/modify blood viscosity [34,35] (Figure 1).

## 3. PART TWO: Effect of Long-Haul Flights on Physical Performance

Systematic [36,37] and narrative reviews [2,3,21] aimed to summarize the most recent findings and provide practical guidelines on how to best minimize (handle/tackle) the consequences of the air-travel-induced circadian rhythm desynchronization (e.g., jet lag) on various physical performance indicators. Briefly, crossing multiple time zones via air travel and within a limited time frame is instrumental to inducing the loss of synchronicity among the circadian rhythms, thereby affecting sleep, core temperature, gastrointestinal function, and melatonin release, each of which can translate into impaired physical performance in athletes [3].

### 3.1. Fluid Intake and Hydration Status of Athletes on Long-Haul Flights

There are only a limited number of investigations that have analyzed the fluid intake or hydration status of athletes during transmeridian air travel. One study on kite surfers used a questionnaire-based approach to self-report water intake among this population of internationally competing athletes [13], as reviewed previously. A more objective study by Schumacher and co-authors [28] examined the effects of an 8 h long-haul flight from Europe to the Middle East on hematological indices of hydration status in 15 endurance-trained athletes. They found that hemoglobin concentration was slightly more elevated before than after traveling in athletes (+0.5 g/dL, *p* = 0.038), and a similar pattern was noted three days after the athletes had reached their destination. However, there were no differences observed in the hematological variables between athletes and the nontraveling controls. Thus, Schumacher concluded that transmeridian air travel does not influence overall fluid balance and hydration status among those athletes. And yet, Schumacher and co-authors [28] did not measure any urinary indices of hydration status in their work (Table 1).

Urinary markers (e.g., urine specific gravity (USG) in particular) have been suggested by Hamouti et al. [38] to be an adequate index of hydration status compared to other methods, such as using a color chart or analyzing blood markers [38]. Briefly, under well-controlled conditions [38], USG is argued to be a superior dehydration marker (at lower levels of dehydration, ~2%), compared to blood indices primarily due to the body fluid regulatory homeostatic mechanisms, including plasma volume defense pathways [39]. Having said this, there remains no industry gold standard for assessing hydration standards [40,41]. More specifically, because a large portion of fluid remains trapped in plasma following fluid consumption, and after exercise-induced dehydration remains, this state can “trick” the antidiuretic hormone (ADH)-osmoreceptor feedback system, leading to the incomplete fluid restoration of interstitial compartments [39], especially in athletes [42]. Subsequently, an athlete’s thirst sensation and fluid retention can be attenuated, which is then translated into an overall suboptimal fluid restoration [42]. To address some of the unresolved issues, Cotter et al. [35] conducted a double-blind, placebo-controlled, crossover study which examined the effects of a commercially available beverage during a 7 h long transmeridian flight and its effects on the hydration status of 12 healthy volunteers. Hydration status was measured via bioimpedance and urinary output measures. Their findings indicated that consuming 330–400 mL of the commercially available drink during the long-haul transmeridian flight significantly lowered urine output [1.05 (0.48) vs. 1.28 (0.34) L, mean (CI)] and increased plasma volume by ~4% (bioimpedance) compared to the placebo drink.

Although the above-mentioned studies have attempted to analyze the effects of long-distance air travel on hydration status, and many studies did evaluate the potential influence of dehydration on physical performance or its associated physiological mechanisms, there is no clear study investigating these effects in a consistent manner. Indeed, the importance of utilizing a proper hydration strategy was recently discussed by van Rensburg et al. [36], who examined the effects of jet lag on physical performance in athletes in their systematic review. They looked at a number of factors and concluded that remarkably, there are still no studies looking at the combined effects of hydration status on physical performance of athletes traveling across multiple time zones.

### 3.2. Effects of Caffeine Consumption on Hydration Status during Long-Haul Flights

To overcome the acute, jet-lag-induced symptoms of fatigue and potential declines in physical performance, caffeine consumption during transmeridian flights is a widely accepted strategy [13,43,44]. Current guidelines from the European Food Safety Authority conclude that caffeine ingestion (of up to 6 mg kg^−1^ body weight) will not induce diuresis; however, their conclusion is only related to caffeine ingestion coupled with endurance exercise involvement. To consider these points, Seal et al. [44] studied the effects of different caffeine dosage on fluid balance and hydration status. Subjects completed three trials on separate occasions at least five days apart in a counterbalanced, crossover study design. All participants consumed 269 ± 45 and 537 ± 89 mg of caffeine for LOW-CAF and HIGH-CAF trials, respectively, representing a low and high dose of caffeine. Urine was collected at 60, 120, and 180 min after the test drink ingestion and urine volume was measured. The data indicate that caffeine intake of 6 mg kg^−1^ in the form of coffee can induce an acute diuretic effect, while 3 mg kg^−1^ does not disturb fluid balance in healthy casual coffee-drinking adults at rest.

These findings are in line with original work reported by Killer et al. [45] and a meta-analysis [46] where authors demonstrated that the substantial fluid loss associated with lower concentrations of caffeine consumption is unjustified, especially when caffeine is consumed prior to exercise. Specifically, Killer et al. [45] used a counterbalanced crossover design in 50 male coffee drinkers who habitually consumed 3–6 cups per day. They participated in two trials, each lasting three consecutive days. During the study, their physical activity, food, and fluid intake were controlled; participants consumed either 4 × 200 mL of coffee containing 4 mg/kg caffeine (C) or water (W) [47]. Deuterium oxide was used to assess the total body water (TBW) pre- and post-trial via ingestion. Urinary and hematological hydration markers were recorded daily in addition to nude body mass measurement (BM). Plasma was analyzed for caffeine to confirm compliance. Their findings showed no significant changes in TBW from beginning to end of either trial and no differences between trials (51.5 ± 1.4 vs. 51.4 ± 1.3 kg, for C and W, respectively). No differences were observed between trials across any hematological markers or in 24 h urine volume (2409 ± 660 vs. 2428 ± 669 mL, for C and W, respectively), USG, osmolality, or creatinine. Mean urinary Na (+) excretion was higher in C than W (*p* = 0.02). Collectively, data by Killer et al. [47] suggest that coffee, when consumed in moderation by caffeine-habituated males, provides similar hydrating qualities to water. Finally, Zhang et al. [46] summarized that caffeine ingestion (~300 mg) was translated into a minor diuretic effect; these were negated by exercise. Concerns regarding unwanted fluid loss associated with caffeine consumption might be unjustified, particularly when ingestion precedes exercise. The caffeine ingested was not corrected for body mass, and both men and women were included in this work, making it difficult to compare with other literature, since data on anthropometrics were not reported.

### 3.3. Hypohydration, Jet Lag, and Performance

It is still unknown whether hypohydration per se accelerates declines in physical performance or imposes additional fatigue development following transmeridian air travel, since rapidly crossing multiple time zones is known to impair physical performance. Jet lag side effects and subsequent performance outcome parameters that have been assessed are diverse, including jump performance [48,49,50], sprint performance [48,49], isometric adductor strength [47], and handgrip strength [51,52], all presented in Table 2. Performance decrements were predominantly observed throughout the explosive movement of the lower limbs (e.g., countermovement jump decreases immediately upon arrival by 7–10%, and a similar pattern was observed in sprint performance), but not during isometric contractions of the lower or upper limbs. The mechanisms governing neuromuscular function after long-haul travel remain unknown, since the above-mentioned measures nearly entirely use gross estimates of neuromuscular function. More comprehensive insight via EMG or interpolated twitch techniques is still needed. Interestingly, there are as yet no data on the combined effects of jet lag and dehydration on neuromuscular performance in weight-class athletes, including combat sports athletes, weightlifters, gymnasts, or sailors [53,54,55,56]. This is rather unexpected, since it is well known that these athletes will typically restrict their food and fluid intake ahead of competition to maintain lower body weight in an attempt to increase their likelihood of competitive success. These athletes typically have suboptimal fluid intake [53], and food and fluid deprivation are known to negatively affect neuromuscular performance in combat sport athletes (e.g., Olympic-style boxers) [55], and sailors also tend to dehydrate during competition [56].

In terms of the cardiorespiratory fitness and the cardiovascular response to long-distance air travel, data on maximal oxygen uptake and/or oxygen uptake kinetics in elite athletes are sparse. One study evaluated the effects of 10 h airline travel on oxygen saturation (SpO_2_, %) and heart rate (HR) in 45 national-level athletes [57]. They found a significant acute decrease in the SpO_2_ (by ~4%) during the transmeridian air travel with readings returning to baseline values within 7 h after landing; HR was unaffected throughout. Montaruli et al. [59] studied the effects of transmeridian air travel from Italy to the USA in marathon runners. These runners were further separated into two experimental and one control group to examine whether a training schedule (i.e., time of the day) affected sleep patterns (assessed via actigraphy). They found sleep quality was mostly preserved with a preplanned training routine, but unfortunately, they failed to report any data on the physiological profiles of these runners, or their performance outcomes at the New York marathon. Lemmer et al. [51] evaluated the HR rhythms and blood pressure (BP) response in elite German athletes traveling both eastward and westward to compete internationally. In brief, they found that the HR and BP were not affected immediately after arrival at their destination, but on day two after arrival, BP increased in those athletes traveling westwards, and decreased in those traveling eastwards. Similar adjustments to HR were noted, lasting up until day 11 for those athletes traveling westward. Lemmer et al. [51] also reported the negative influence of transmeridian air travel on core temperature fluctuations and saliva melatonin and cortisol concentrations, but again no data were shown on the hydration status of the participants included or the potential influence of acute dehydration on any of the above-mentioned physiological indicators.

The hormonal response and inflammation in athletes traveling and competing across different time zones are often evaluated across investigations [49,51,52]. Salivary cortisol and melatonin concentrations are each affected by transmeridian travel, and apparently, to a greater extent for athletes traveling eastward [51]. Similar findings have been observed by Bullock et al. [49] who reported a ~67% lower resting salivary cortisol level immediately after reaching their competition destination, whereas no changes were observed in the nontraveling control group. Contrary to the above-mentioned, recent work of Kraemer et al. [52] reported that plasma epinephrine, testosterone, and cortisol concentrations are significantly elevated above baseline before and after a stimulated competition, irrespective of the actual 6 h long trans-American air travel time. Interestingly, none of these investigations explored the potential influence of hydration status on the hormonal disturbance following long-haul air travel.

Finally, subjective, self-perceived data on sleep quality, overall fatigue, muscle soreness, and mental preparedness uniformly suggest an impaired physical capacity in athletes immediately after the long-haul flights which include large time zone changes [13,47,49,57]. Unfortunately, data from these subjectively measured scales have not been combined with data on actual fluid intake, self-perceived thirst scales, or USG readings in any of the above-mentioned papers. A possible explanation for the lack of data on the combined effects of jet lag and acute dehydration may be primarily from a methodological perspective. It seems rather challenging to isolate the independent and combined effects of acute hypohydration from other long-haul, transmeridian, travel-induced symptoms and side effects. This is especially true when no study has evaluated the hydration status of its traveling athletes despite significant evidence that the air cabin environment creates an environment promoting fluid loss of the individual by the very nature of the air quality, composition, and sedentary physical inaction of air travel. Thus, both structural and functional aspects of human physiology are affected by long-haul flights and the air cabin environment but to what extent dehydration is a risk factor for traveling athletes remains to be determined.

## 4. Future Directions and Practical Recommendations

Despite the lack of studies directly investigating how modulating hydration status may prevent or reduce jet lag symptoms, some recommendations may be proposed based on preliminary findings and translating results of other research in this field.

Good hydration should be an everyday practice; be sure to start travel in an optimally hydrated state. General recommendations for everyday fluid balance may differ when applied to athletes.During long-haul flights, fluid intake should be between 100 and 300 mL/h (including that which is derived from food). These values are based on estimated fluid losses [4] and from preliminary research findings [11,35]. Adapting one’s fluid intake timing in order to avoid disruption during sleep is highly recommended.Hydration strategies which incorporate electrolytes paired with carbohydrate solutions may be beneficial, in addition to pure water [11,35].If not habituated, avoid caffeine intake since it may (i) increase diuresis and (ii) impair sleeping. If habituated, a total caffeine ingestion ≤300 mg has been shown to have minimal adverse effects on travelers [44].

Future studies should focus on determining exact changes in hydration status across long-haul flights and juxtapose these with severity of jet lag symptoms. Only until the magnitude of the dehydrating effect of flying is quantified will it then be possible to design appropriate guidelines and countermeasures for travelers. In particular, our analysis of the literature has highlighted some questions and issues that may warrant further investigation, such as (i) what is a common consensus on increased fluid losses during long-haul flights (assuming similar cabin conditions), (ii) determining prevalence of hypo/underhydration during long-haul flights and its relationship with the severity of jet lag symptoms, and (iii) the need for more placebo-controlled, double-blind studies to determine the efficacy of nutrition/hydration protocols which can be used to maintain fluid balance and mitigate jet leg symptoms.

## 5. Conclusions

There is a lack of randomized controlled trials exploring the effects of acute hydration status on various physiological performance indicators following long-distance air travel. After reviewing the limited studies which investigated any physiological variable on actual traveling athletes, there is no consensus or real evidence of how the incidence rate, magnitude, or duration of acute dehydration due to air travel affects the general health or performance of elite athletes. It remains that the air cabin environment does provide a situation where significant changes in fluid balance may occur.

## Figures and Tables

**Figure 1 nutrients-12-02574-f001:**
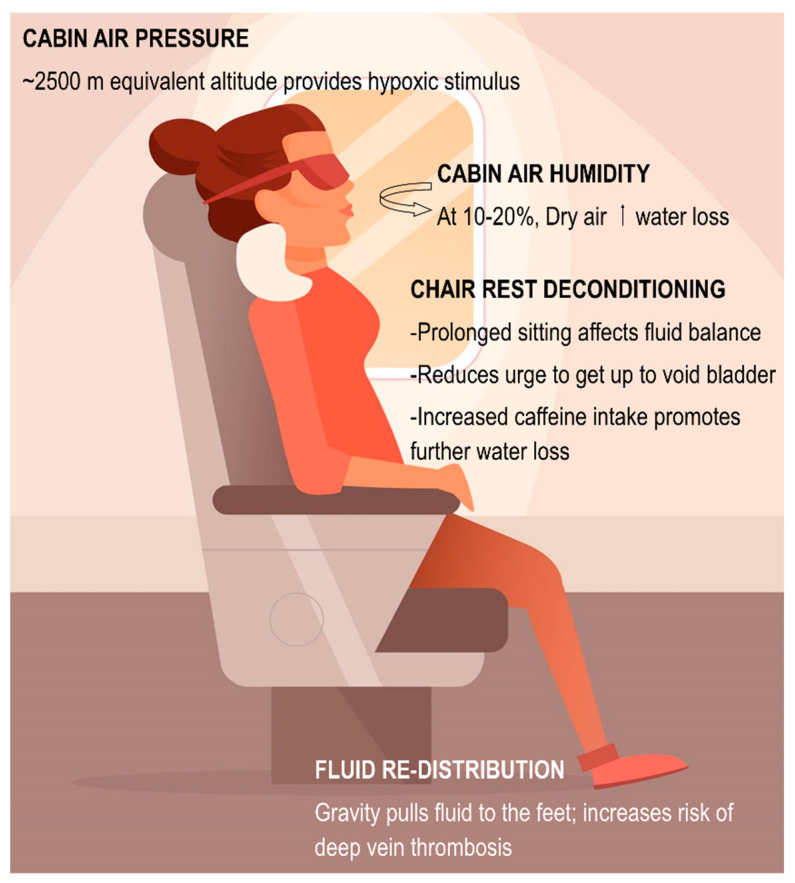
Air cabin environment creates a unique situation that promotes possible increased risk for dehydration and fluid shift en route to the destination which may be further exacerbated by the behavior of the individual and independent of jet-lag-induced alterations in circadian rhythm.

**Table 1 nutrients-12-02574-t001:** Factors influencing fluid balance during long-haul flights.

Factor	Expected Changes	Measured Changes	E.g. London–Tokyo (~12 h)
Insensible and ventilatory fluid losses	Increased due to reduced air humidity and slightly increased ventilation	Increased by 200 mL/h (360 mL/h total) [4]	Increased fluid consumption by at least 2.4 L (up to 4 L)
Gastrointestinal losses	Altered motility and absorption due to circadian rhythm alterations and dietary patterns may affect both fluid intake and fluid losses	Abdominal pain, constipation, and diarrhea are often reported [19]. No clear data on changes in food and fluid intake or fluid losses	Diarrhea or constipation may increase or decrease fluid losses, respectively
Urinary losses	Circadian rhythm alterations may influence kidney function	No clear studies are present, possible hyperfiltration and microalbuminuria [24,25]	No data are available to make a clear estimation
Fluid intake	Reduced fluid intake may be expected due to altered appetite, gastrointestinal dysfunction, changes in dietary habits, and desire to reduce the need to void the bladder	Few studies assessed fluid intake during long-haul flights in athletes. Less than 500 mL may be consumed for long-distance travel [13]	Fluid intake of minimum 200–250 mL/h may be encouraged (consider fluid from food)

**Note:** Due to the lack of a common consensus on total fluid losses during long-haul flights, caution should be applied when approximating/estimating fluid needs.

**Table 2 nutrients-12-02574-t002:** Summary of the physiological effects of jet lag and transmeridian travel on actual athletes traveling across multiple time zones.

Study	Participants	Population	Flight Details	Design	Testing Methodology	Performance Outcomes	Fluid Intake?	Hydration Status Assessed?
Chapman et al. [48]	*N* = 5 experimental group (4 female) *N* = 7 control (6 female) Age: 25 ± 6 y	Skeleton athletes Australia and Canada national team members	AUS to CAN 8 time zones WEST TT: ~24 h	Cross-sectional cohort Travel team (AUS) compared to nontravel team (CAN)	Data collection: 2 d before, immediately after travel, and 6 times in the 10 d postflight duration Lower body power tests (power and velocity)	Dec in peak and mean squat jump velocity; CMJ velocity did not change; CMJ jump height decreased; squat movement NS	Not reported	No
Bullock et al. [49]	Same as above	Same as above	Same as above	Same as above	Test sampling as above 30 m sprint performance	NS for performance time in travel group; saliva cortisol decreased 67%; no change in nontravel group	Not reported	USG measured; NS on any postflight measurement day; NS different from nontravel group
Broatch et al. [50]	*N* = 12 females 25 ± 2 y	Volleyball	Canberra, AUS to Manila, Philippines 2 time zones NORTHWEST TT: 6.5 h	RCT compression garments (randomized design, *n* = 6 per group)	Data collection 1 d before, 12, 24, and 48 h postflight, resting BP and HR during flight	CMJ; NS time or interaction effect; mean velocity higher for compression group by 4% 24 h post; relative power 8% higher 24 h post; resting BP and HR NS main effects; SaO_2_ less attenuated with compression at 6.5 and 9 h postflight; calf girth NS; markers of blood clotting NS	Not reported	No
Fowler et al. [47]	*N* = 18 males 24 ± 3 y	Professional AUS Rugby League World series 2015	AUS to UK 11 time zones WEST TT: 24 h	Cross-sectional pre/post	Data collected: 1 d before, +2, +6, and +8 d post-travel Muscle function (isometric force via adductor squeeze dynamometry, flexibility, and ROM) Self-reported: sleep, jet lag questionnaire	NS across any dependent measure at any time-point except self-reported upper respiratory symptoms 6 d post-travel in *n* = 6 (30%) of athletes	Not reported	No
Lemmer et al. [51]	WEST *N* = 13 male 25 ± 2 y EAST *N* = 6 male 23 ± 2 *N* = 4 athletes participated in both directions Control data collection on same subjects	Elite gymnasts	Frankfurt, Germany to Atlanta, USA 6 time zones WEST TT: 10 h AND Munich, Germany to Osaka, Japan 8 time zones EAST TT:12 h	Cross-sectional with follow-up assessments	Data collected: As control in Germany, then +1, 4, 6, and 11 d after arrival to destination for each direction 24 h before HR, BP; oral temperature; handgrip strength; saliva for cortisol and melatonin Self-reported: jet lag symptoms	All functions were disturbed on the first day on arrival at destination (both directions) and remained until 5–6 d (WEST) and 7 d after EAST BP and HR affected on first day also interaction effect *WEST* SBP inc. ~5–8 mmHg DBP no sig. changeHR inc ~16 bpm *EAST* SBP dec ~11 mmHg DBP dec ~7 mmHg HR no sig. change Handgrip not affected in either direction	Not reported	No
Kraemer et al., [52]	*N* = 10 athletes (compression) 23 ± 2 y and *N* = 9 athletes (matched controls) 23 ± 2 y	“Recreational” athletes, unspecified	Hartford CT, USA to Los Angeles, CA, USA (return) 3 time zones WEST and EAST 5–6 h total travel time each direction	RCT Compression garments (worn on upper and lower body for entire testing time, incl. sleeping)	Data collected: 6 total test days 1 d before (pm) (west-east) and on arrival, Day 2 pre-/postexercise testing, +1 and +2 days after return flight Blood + urine sampling: markers of inflammation and hormonal disturbance Physical performance indicators: handgrip strength, CMJ, 40-yard sprint, pro-agility drills Self-reported: jet lag questionnaire	Compression garments maintained lower body indicators; no changes in isometric strength of upper limbs	Not reported Methodology specified that if USG was >1020, athletes were instructed to “drink water” until USG was <1020	Yes USG measured throughout, but results not reported or correlated to any performance outcomes.
Geertsema et al., [57]	*N* = 48 athletes and *N* = 18 controls Sex not reported 26 ± 3 y	“Athletes” not specified. In-text defined as “involved at the senior level in an aerobic sport”	Pooled data from *N* = 10 flights Range: 2 flights 3h and 8 flights >10 h	Prospective cross-sectional	Data collected: preflight 3 h, 7 h during flight, at destination (in the airport); HR and O_2_ saturation	NS HR O_2_ decreased 4% inflight, and returned to baseline immediately upon arrival	Not reported	No
Schumacher et al., [28]	*N*= 15 male 28 ± 4 y *N* = 11 male controls 25 ± 3 y	Endurance athletes	Germany to Doha, Qatar 2 time zones (EAST) TT: 14 h	Prospective cross-sectional	Data collected: 1 d before (am) +1 d on arrival (am) +3 d after (am and pm); Bloodwork	None per se [Hb] and [Hct] were each sig. lower on +1 d am sample, but no different between groups and resolved by +3 d post No correlation between blood markers or change in weight on travel day	Yes Fluid intake recorded on the travel day (2750 ± 521 mL) of “different” beverages consumed Urine output: 1768 ± 621 mL No change in weight	Indirectly yes Changes in weight and overall fluid intake were measured during travel. Blood markers were assessed, but plasma volume not calculated from these variables.
Stevens et al., [58]	*N* = 12 male 48 ± 14 y	Triathletes (Masters) Hawaii Ironman	Sydney, AUS to Kona via Honalulu, USA NORTHEAST Flight time 12 h TT: 23 h,	Prospective cohort	Data collected: 10 d, 7 d, and 4 d preflight, 2 d, 3 d, 4 d, 1 d before race day, during race, 1 d after race Sleep quality (actigraph) Saliva: (sIgA, sCort) Self-reported: sleep diary, illness, jet lag, physical preparedness	No changes in sleep quality or mucosal measures across the study	No	No

Notes: CMJ, countermovement jump; TT: total time traveled; NS, not significant; RCT, randomized control trial; HR, heart rate; BP, blood pressure; SBP, systolic blood pressure; DBP, diastolic blood pressure; USG, urine specific gravity.

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
