# Peer review of "Up in the Air: Evidence of Dehydration Risk and Long-Haul Flight on Athletic Performance"

_nutrients, 2020, doi:10.3390/nu12092574_

Round 1
Reviewer 1 Report
This review presents an elegant look at a common conception in the field of air travel and hydration, with particularly relevant concerns for athletes.
Part 1:
The mechanistic review is adequate, but could be extended one step further to model out typical expected fluid losses. Consider including this as a table.
Part 2:
I would recommend changing the header to better reflect this section's hydration centric view of the performance consequences of flight.
Line 257-Line 258: This conclusion does not seem to directly align with the above stated section. It would appear to indicate the authors are proposing that caffeine does exert an effect on hydration status. Consider rewording.
Section 4: Given the paucity of data presented in the article, I'm not sure there is a great background to support these recommendations. While they appear to be "common sense", it seems contrary to the rest of the manuscript. Consider removing this section.
Conclusion: I would caution the use of the phrase "significant changes in fluid balance". Through convention in the field this would suggest >2% body mass losses. I don't feel that is supported by the presented data. Consider revision.
Author Response
Manuscript ID: nutrients-883626, entitled: Up in the air: evidence of dehydration risk and long-haul flight on athletic performance
We are very grateful for the opportunity to revise our manuscript, and many thanks to the reviewers for their very helpful comments. We have carefully updated our manuscript, and we appreciate your efforts to further improve our work. We feel that several important points in our previous submission were correctly identified in need of revision, and as a result of these revisions, the manuscript is now a much stronger and clearer article. All corrections are marked with green color below. We hope that these revisions and our accompanying responses will be sufficient for our manuscript to be deemed suitable for publication in the Nutrients journal.
Reviewer: 1
Comments to the Author:
This review presents an elegant look at a common conception in the field of air travel and hydration, with particularly relevant concerns for athletes.
Response: Thank you for recognizing the potential of our work. We have tried to meet all of your suggestions. You have raised several important issues, so please see how we dealt with each one of your comments, line by line.
Line by line:
Part 1:
The mechanistic review is adequate, but could be extended one step further to model out typical expected fluid losses. Consider including this as a table.
Response: Thank you for input and suggestion. We have included a new “Table 1. Factors influencing fluid balance during long-haul flights” to model out possible expected fluid losses from various factors. The table summarizing fluid balance factors that may be altered during long-haul flights, and we simulated how much fluid intake is expected to be needed during a flight from London to Tokyo as an example for the (planned) next Olympic games.
Part 2:
I would recommend changing the header to better reflect this section's hydration centric view of the performance consequences of flight.
Response: Thank you for your suggestion, we have slightly re-worded. The MS reads now:
“Mechanisms of Hydration State Changes During Long-Haul Flights”
Line 257-Line 258: This conclusion does not seem to directly align with the above stated section. It would appear to indicate the authors are proposing that caffeine does exert an effect on hydration status. Consider rewording.
Response: Good catch, we agree, therefore this sentence was removed from the MS.
Section 4: Given the paucity of data presented in the article, I'm not sure there is a great background to support these recommendations. While they appear to be "common sense", it seems contrary to the rest of the manuscript. Consider removing this section.
Response: Thank you for your suggestion, we have followed your input and removed a large portion of recommendations from the MS. We have kept only those which were supported by the literature. We have directed this manuscript to be helpful not only to academics, but also (hopefully) as a useful message for both athletes and coaches, to understand that there are possible risks of dehydration during long-haul flights, so they might make possible countermeasures. Therefore, in line with your comments and those of the other reviewer, we have modified this section to include not just a few guidelines based on the literature we know, but also on future directions that researchers may choose to investigate further.
Conclusion: I would caution the use of the phrase "significant changes in fluid balance". Through convention in the field this would suggest >2% body mass losses. I don't feel that is supported by the presented data. Consider revision.
Response: We agree, therefore the MS was slightly re-worded, it reads now: “It remains that the air cabin environment does provide a situation where changes in hydration status may occur”.
Reviewer 2 Report
General Comments: This review does a nice job of filling in gaps in the literature based and providing practical applications for athletes during flights. Overall, the flow of the manuscript is good, however, the numbering of the various sections is not clear. Instead of using 2. PART 1 and 2. PART 2 would it be possible to simply change those to 2 and 3? This review could also benefit from a section that describes why dehydration from flying is detrimental to athlete performance. Consider integrating potential future research opportunities into this manuscript. An additional paragraph or statements at the end of specific paragraphs may help guide readers to what is needed in this area.
Abstract: Consider adding a sentence at the end of the abstract that summarizes the recommendations you are giving in this manuscript.
- Study Rationale: Consider adding a clear statement about why information in this review article is needed and/or what the reader will gain from this manuscript.
Line 30-33: Please provide citations for the reviews mentioned here.
Line 34: The title of this section isn’t completely clear. Consider revising to Mechanisms of Hydration State Changes During Long-Haul Flights
Line 57-59: Please clarify this statement. Is this suggesting that in flight, water losses are at least 360 mL/hour so the previously recommended 15-20 mL is vastly under prescribing?
Line 96-106: This paragraph does not seem to be focused on hydration. Consider relating this background information to what implications this will have on overall fluid balance.
Line 116: A recommendation about what athletes could possibly do about this problem might be useful at the end of this paragraph.
Line 142: One limitation to this study was that urinary hydration markers were not assessed so a complete picture of hydration status was unavailable. Consider adding that statement to this paragraph.
Line 165-168: This statement could be moved to the study rationale section.
Line 189-191: This reviewer disagrees with the statement that USG is a superior index of hydration. Please consider citing the following manuscripts and revise this paragraph accordingly. Assessing hydration: the elusive gold standard. L.E. Armstrong. J Am Coll Nutr. 2007 Oct; 26(5 Suppl):575S-584S and Assessing hydration status. S.A. Kavouras. Curr Opin Clin Nutr Metab Care. 2002 Sep;5(5):519-24.
Line 203-204: Please describe how “better preserved fluid balance” was defined. Was this with Bioimpedance or urinary output measures?
Line 205-207: Consider removing the Kraemer study since they did not assess hydration status during or after the flight.
Line 223-224: Please cite the general flying guidelines that are mentioned here.
Line 257-258: This statement might be misleading because it makes it seem like some caffeine is slightly bad but more caffeine is worse. As stated above in the same paragraph, it doesn’t seem that caffeine in moderation has any impact on overall hydration state. Consider revising this sentence to reflect that conclusion.
Line 329-330: Consider removing the guidelines for 2.5 L and 2.0 L for males and females as this is most likely not sufficient for athletes who are training almost daily. Instead, consider suggesting the use of the WUT Venn diagram to ensure adequate hydration.
Line 341-343: Consider removing this recommendation as this review did not cover this topic extensively.
Line 344: See above.
Table 1:
If possible, consider putting flight details in the same format.
Author Response
Manuscript ID: nutrients-883626, entitled: Up in the air: evidence of dehydration risk and long-haul flight on athletic performance
We are very grateful for the opportunity to revise our manuscript, and many thanks to the reviewers for their very helpful comments. We have carefully updated our manuscript, and we appreciate your efforts to further improve our work. We feel that several important points in our previous submission were correctly identified in need of revision, and as a result of these revisions, the manuscript is now a much stronger and clearer article. All corrections are marked with green color below. We hope that these revisions and our accompanying responses will be sufficient for our manuscript to be deemed suitable for publication in the Nutrients journal.
Reviewer: 2
Comments to the Author
This review does a nice job of filling in gaps in the literature based and providing practical applications for athletes during flights. Overall, the flow of the manuscript is good, however, the numbering of the various sections is not clear. Instead of using 2. PART 1 and 2. PART 2 would it be possible to simply change those to 2 and 3? This review could also benefit from a section that describes why dehydration from flying is detrimental to athlete performance. Consider integrating potential future research opportunities into this manuscript. An additional paragraph or statements at the end of specific paragraphs may help guide readers to what is needed in this area.
Response: Thank you very much for your time in reviewing the manuscript and your thoughtful suggestions throughout the work. We tried to address each point in the following portion of the text, please see how we dealt with each issue raised, below - line by line. We changed the labeling of PART 1 and PART 2 to so that the numbers align and hopefully this helps the reader with clarity. We integrated future research opportunities at a new section at the end of the paper after considering both yours and the other reviewer comments.
Line by line:
Abstract: Consider adding a sentence at the end of the abstract that summarizes the recommendations you are giving in this manuscript.
Response: Thank you for your input, we have added the following sentence: “Upon review, there are few studies which have been conducted on actual travelling athletes, those that do provide no real evidence of the incidence rate, magnitude, or duration acute dehydration may pose on the general health or performance of elite athletes.”
Study Rationale: Consider adding a clear statement about why information in this review article is needed and/or what the reader will gain from this manuscript.
Response: Thank you for your thoughtful suggestion, this portion of the MS was amended, it reads now:
“Every major review article on jetlag over the past ~10 years have suggested avoiding dehydration or advocating for staying hydrated whilst flying, although none of the works discuss in detail the magnitude of the expected levels of dehydration to be experienced over a given flight duration, nor whether athletes could be at greater risk for becoming hypohydrated. To compete internationally, elite athletes are frequently exposed to long-distance air travel across multiple time zones. This results in a cluster of acute detrimental health-related side effects and symptoms, commonly known as jetlag. Thus, there is growing interest to determine the influence trans-meridian air travel has on the physical performance of athletes.”
Line 30-33: Please provide citations for the reviews mentioned here.
Response: Thank you for you input, these references are now included in the MS.
Line 34: The title of this section isn’t completely clear. Consider revising to Mechanisms of Hydration State Changes During Long-Haul Flights
Response: Thank you for your suggestion, we have amended according to the suggestions of the other reviewer. Please see line section Title, line 34-35.
Line 57-59: Please clarify this statement. Is this suggesting that in flight, water losses are at least 360 mL/hour so the previously recommended 15-20 mL is vastly under prescribing?
Response: We thank the reviewer for this suggestion. The sentence was poor worded, so we rephrased it. Yes, you are correct that previous research is indeed vastly under prescribing possible water losses in this environment.
Line 96-106: This paragraph does not seem to be focused on hydration. Consider relating this background information to what implications this will have on overall fluid balance.
Response: A better transition between paragraphs has been added: “Suffering from these side-effects would have a direct influence on fluid intake behavior and overall fluid balance of any traveler. If nutrient absorption and gastric and gut motility are affected by circadian disruption, then it may be that the athlete cannot absorb as much water as is needed to remain well hydrated, or they may simply lack the appetite to consume additional liquids due to actual gut discomfort.”
Line 116: A recommendation about what athletes could possibly do about this problem might be useful at the end of this paragraph.
Response: We added a concluding sentence to the paragraph and the new Table 1 which should help satisfy the reader with your comment above.
Line 142: One limitation to this study was that urinary hydration markers were not assessed so a complete picture of hydration status was unavailable. Consider adding that statement to this paragraph.
Response: We agree, this statement was added to the MS. Please see line 158-160.
Line 165-168: This statement could be moved to the study rationale section.
Response: We agree; this statement was moved according to your suggestion.
Line 189-191: This reviewer disagrees with the statement that USG is a superior index of hydration. Please consider citing the following manuscripts and revise this paragraph accordingly. Assessing hydration: the elusive gold standard. L.E. Armstrong. J Am Coll Nutr. 2007 Oct; 26(5 Suppl):575S-584S and Assessing hydration status. S.A. Kavouras. Curr Opin Clin Nutr Metab Care. 2002 Sep;5(5):519-24.
Response: Thank you for your suggestion, our statement was amended accordingly and the above-mentioned references were included as well.
Line 203-204: Please describe how “better preserved fluid balance” was defined. Was this with Bioimpedance or urinary output measures?
Response: A more detailed account of the authors’ findings now reads as: “Their findings indicated that consuming 330 - 400 mL of the commercially available drink during the long haul trans-meridian flight significantly lowered urine output [1.05 (0.48) vs. 1.28 (0.34) L, mean (CI)] and increased plasma volume by ~4% (bioimpedance) compared to the placebo drink.” We hope this clarifies the issue.
Line 205-207: Consider removing the Kraemer study since they did not assess hydration status during or after the flight.
Response: Thank you for your suggestion, this references was removed from the MS.
Line 223-224: Please cite the general flying guidelines that are mentioned here.
Response: This was a loosely-written sentence which did not contribute to the understanding of the paragraph and has thus been removed.
Line 257-258: This statement might be misleading because it makes it seem like some caffeine is slightly bad but more caffeine is worse. As stated above in the same paragraph, it doesn’t seem that caffeine in moderation has any impact on overall hydration state. Consider revising this sentence to reflect that conclusion.
Response: We agree, therefore this sentence was removed from the MS.
Line 329-330: Consider removing the guidelines for 2.5 L and 2.0 L for males and females as this is most likely not sufficient for athletes who are training almost daily. Instead, consider suggesting the use of the WUT Venn diagram to ensure adequate hydration.
Response: Thank you for this suggestion. We included a more individualized hydration status assessment, citing the urine colour chart. We removed the specific guidelines to avoid confusion.
Line 341-343: Consider removing this recommendation as this review did not cover this topic extensively.
Response: We agree; therefore, this statement was removed.
Line 344: See above.
Response: We agree; removed.
Table 1: If possible, consider putting flight details in the same format.
Response: Flight details in the table appear in the following order (a) Travel to/from destination (b) direction (c) TT travel time. If there are entries that do not conform to this format, it means we were unable to find that information in the article.
Once again, we thank both reviewers for their great efforts, valuable input and willingness to make this manuscript even stronger.

Round 2
Reviewer 2 Report
Thank you for updating the manuscript according to review comments. I have no further suggestions.